# Computing Higher Order Derivatives of Matrix and Tensor Expressions

**Sören Laue**
Friedrich-Schiller-Universität Jena
Germany
`soeren.laue@uni-jena.de`

**Matthias Mitterreiter**
Friedrich-Schiller-Universität Jena
Germany
`matthias.mitterreiter@uni-jena.de`

**Joachim Giesen**
Friedrich-Schiller-Universität Jena
Germany
`joachim.giesen@uni-jena.de`

## Abstract

Optimization is an integral part of most machine learning systems and most numerical optimization schemes rely on the computation of derivatives. Therefore, frameworks for computing derivatives are an active area of machine learning research. Surprisingly, as of yet, no existing framework is capable of computing higher order matrix and tensor derivatives directly. Here, we close this fundamental gap and present an algorithmic framework for computing matrix and tensor derivatives that extends seamlessly to higher order derivatives. The framework can be used for symbolic as well as for forward and reverse mode automatic differentiation. Experiments show a speedup of up to two orders of magnitude over state-of-the-art frameworks when evaluating higher order derivatives on CPUs and a speedup of about three orders of magnitude on GPUs.

## 1 Introduction

Recently, automatic differentiation has become popular in the machine learning community due to its genericity, flexibility, and efficiency. Automatic differentiation lies at the core of most deep learning frameworks and has made deep learning widely accessible. In principle, automatic differentiation can be used for differentiating any code. In practice, however, it is primarily targeted at scalar valued functions. Current algorithms and implementations do not produce very efficient code when computing Jacobians or Hessians that, for instance, arise in the context of constrained optimization problems. A simple yet instructive example is the function $f(x) = x^\top A x$, where $A$ is a square matrix and $x$ is a vector. The second order derivative of this function, i.e., its Hessian, is the matrix $A^\top + A$. Frameworks like TensorFlow [1], Theano [23], PyTorch [16], or HIPS autograd [14] generate code for the second order derivative of $f$ that runs two to three orders of magnitude slower than the evaluation of the expression $A^\top + A$.

In machine learning, automatic differentiation is mostly used for numerical optimization. Many machine learning optimization problems are formulated in standard matrix language. This has not only the advantage of a compact problem representation, but linear algebra expressions can also be executed efficiently on modern CPUs and GPUs by mapping them onto highly tuned BLAS (basic linear algebra subprograms) implementations that make extensive use of vectorization such as the SSE and the AVX instruction sets or SIMD architectures. Ideally, these advantages are kept also for the gradients and Hessians of the expressions that encode the optimization problem. This is the purpose of matrix calculus that computes, if possible, derivatives of matrix expressions again as

matrix expressions. In matrix calculus, the gradient of $f(x) = x^\top A x$ is the expression $A^\top x + Ax$ and its Hessian is the expression $A^\top + A$, which can be efficiently evaluated.

Surprisingly, none of the classic computer algebra systems such as Mathematica, Maple, Sage, or SymPy[1] supports matrix calculus. These systems calculate derivatives of a given matrix expression only on matrix entry level, i.e., every matrix entry gives rise to a separate symbolic scalar variable. Each of these systems is able to compute derivatives of univariate scalar-valued functions. Hence, while in pure matrix calculus a matrix $A \in \mathbb{R}^{n \times n}$ is represented by just one variable, the classic computer algebra systems treat this as $n^2$ variables. Thus, the number of input variables increases dramatically and the complexity of evaluating derivatives explodes. Also, it is impossible to read off the derivative in matrix notation from such complex expressions. These drawbacks are also present in the classic frameworks for automatic differentiation that mostly compute derivatives only on scalar level, like ADOL-C [25] or TAPENADE [10]. While the direct integration of matrix and tensor operators into automatic differentiation frameworks like TensorFlow is under active development, so far the output functions still have to be scalar. Hence, higher order derivatives or Jacobians cannot be computed directly.

**Contributions.** We provide an algorithmic framework for computing higher order derivatives of matrix and tensor expressions efficiently, which fully operates on tensors, i.e., all variables are allowed to be tensors of any order, including the output variables. Therefore, for the first time higher order derivatives and Jacobians can be computed directly. The derivatives are represented as compact matrix and tensor expressions which can be mapped to efficient BLAS implementations. Other state-of-the-art frameworks produce huge expression graphs when deriving higher order derivatives. These expression graphs cannot be mapped to simple BLAS calls but involve many for-loops and complex memory access. Hence, we observe an increase in efficiency of up to two orders of magnitude on CPUs. Since GPUs can deal with this even worse we observe here an increase in efficiency of about three orders of magnitude over current state-of-the-art approaches. An important benefit of working on tensors is that it enables symbolic matrix calculus. We show that the predominantly used standard matrix language is not well suited for symbolic matrix calculus, in contrast to a tensor representation. Thus, to implement matrix calculus, we first translate linear algebra matrix expressions into a tensor representation, then compute derivatives in this representation and finally translate the result back into the standard matrix language.

**Related work.** A great overview and many details on fundamentals and more advanced topics of automatic differentiation can be found in the book by Griewank and Walther [9]. Baydin et al. [2] provide an excellent survey on the automatic differentiation methods used within machine learning.

Pearlmutter [17] discusses the following approach for computing higher order derivatives, like the Hessian of a function $f \colon \mathbb{R}^n \to \mathbb{R}$, by automatic differentiation: First, an expression for the gradient $\nabla f$ is computed. Then the Hessian is computed column-wise by multiplying the gradient with the standard basis vectors $e_i$, $i = 1, \dots, n$ and differentiating the resulting $n$ scalar functions $(\nabla f)^\top e_i$. The derivative of $(\nabla f)^\top e_i$ gives the $i$-th column of the Hessian of $f$. Gebremedhin et al. [7] show that a few columns of the Hessian can be computed at once with help of graph coloring algorithms, if the Hessian is sparse and has a special structure. However, if the Hessian has no special structure, then all $n$ columns need to be computed individually. This can be seen, for instance, in TensorFlow's expression graph for computing the Hessian of $x^\top A x$ that has more than one million nodes for a still reasonably small value of $n = 1000$.

An alternative approach is presented in the book by Magnus and Neudecker [15] that provides an extensive number of rules for deriving matrix derivatives. At its core, matrices are turned into vectors by the vec function that stacks the columns of a matrix into one long vector. Then the Kronecker matrix product is used to emulate higher order tensors. This approach works well for computing first order derivatives. However, it is not practicable for computing higher order derivatives. Also, many of the rules rely on the fact that the vec function assumes column-major ordering (Fortran style). If however, one switches to a programming language that uses row-major ordering (C style), then some of the formulas do not hold anymore. For instance, the Python NumPy package that is widely used in the machine learning community follows row-major ordering when used with standard settings. To be independent of such a convention and to be able to compute higher order derivatives, we work directly with higher order tensors.

The work by Giles [8] collects a number of derivatives for matrix operators, i.e., pushforward and pullback functions for automatic differentiation. Similarly, Seeger et al. [22] provide methods and code for computing derivatives for Cholesky factorization, QR decomposition, and symmetric eigenvalue decomposition when seen as matrix operators. However, they all require that the output function is scalar-valued, and hence, cannot be generalized to higher order derivatives.

## 2   Languages for matrix and tensor expressions

Matrix expressions are typically written in a simple yet very effective language. This language features two types of entities, namely objects and operators. The objects of the language are scalars, vectors and matrices, and the operators include addition, various forms of multiplication, transposition, inverses, determinants, traces, and element-wise functions that can be applied to objects.

**Problems with the standard matrix language.** Since matrices can be used to encode different entities like linear maps, bilinear maps, and transformations that describe a change of basis, the matrix language sometimes enforces an indirect encoding of these entities. For instance, let the square matrix $A$ encode a bilinear map on some vector space, i.e., the entries of $A$ represent the evaluation of the bilinear map on any combination of basis vectors. Assume we want to evaluate the bilinear map at the vectors $x$ and $y$ whose entries store the respective coefficients with respect to the same basis that is used for specifying $A$. The evaluation of the bilinear map at $x$ and $y$ is then typically written as $x^\top A y$, which basically means: apply the linear map that is encoded by $A$ to $y$ and feed the resulting vector into the linear form $x^\top$. Note that transposing a vector means mapping it to an element in the dual vector space. The scalar value $x^\top A y$ is not affected by the change of interpretation; the value of the bilinear map evaluated at $x$ and $y$ is the same as the value of the evaluation of the linear form $x^\top$ at the vector $Ay$. Hence, the existence of different interpretations is not a problem when evaluating matrix expressions. But it becomes a problem once we want to compute derivatives of such expressions. For instance, if we want to compute the derivative of the expression $x^\top A x$ with respect to the vector $x$, then we also need to compute the derivative of the transformation that maps $x$ into the linear form $x^\top$, and it is not obvious how to do this. In fact, matrix notation does not contain an expression to represent this derivative.

**Ricci calculus.** The problem is avoided by turning to a different language for encoding matrix expressions, namely Ricci calculus [20]. Ricci calculus lacks the simplicity of the standard language for matrix expressions, but is more precise and can distinguish between linear maps and bilinear maps through the use of indices. In Ricci calculus one distinguishes two types of indices, namely upper (contravariant) and lower (covariant) indices, that determine the behavior of the encoded objects under basis changes. Scalars have no index, vectors have one upper, and covectors one lower index. Bilinear forms on a vector space have two lower indices, bilinear maps on the dual space have two upper indices, and linear maps have one upper and one lower index. Hence, the bilinear map $A$ evaluated at the vectors $x$ and $y$ is written in Ricci calculus as $x^i A_{ij} y^j$, or equivalently $A_{ij} x^i y^j$. Ricci calculus does not include the transposition transformation, but features $\delta$-tensors. The linear identity map is encoded in Ricci calculus by the $\delta$-tensor $\delta^i_j$. The $\delta$-tensors $\delta_{ij}$ and $\delta^{ij}$ have no interpretation as linear maps but serve the purpose of transposition; a vector $x^i$ is mapped by $\delta_{ij}$ to the covector $\delta_{ij} x^i$ and the covector $x_i$ is mapped by $\delta^{ij}$ to the vector $\delta^{ij} x_i$. In Ricci calculus one distinguishes between free and bound indices. Bound indices appear as lower and also as upper indices in the expression, while free indices appear either as lower or upper indices. Hence, the expression $\delta_{ij} x^i$ has one free, lower index, namely $j$, and thus encodes a covector, while $\delta^{ij} x_i$ has one free, upper index, again $j$, and encodes a vector. The expression $A_{ij} x^i x^j$ has no free indices and thus encodes a scalar.

Let us come back to the issue of the interpretation of an expression. The different interpretations of the expression $x^\top A y$ can be distinguished in Ricci calculus; $A_{ij} x^i y^j$ is a bilinear map evaluated at the vectors $x^i$ and $y^j$, and $x_i A^i_j y^j$ is the linear form $x_i$ evaluated at the vector $A^i_j y^j$. The interpretation that the vector $x$ is first mapped to the covector $x^\top$ and then evaluated at the image of the vector $y$ after applying the linear map $A$ is written in Ricci calculus as $x^i \delta_{ij} A^j_k y^k$.

Hence, our approach for computing derivatives of matrix expressions is to translate them first into expressions in Ricci calculus, see Table 1 for examples, and then to compute the derivatives for these expressions. The resulting derivative is again an expression in Ricci calculus that can be translated back into the standard matrix calculus language.

Table 1: Translation of expressions from matrix notation into the Ricci calculus language. Here, $\odot$ denotes the element-wise multiplication and $\mathrm{diag}(x)$ the matrix whose diagonal is the vector $x$.

| Matrix notation | Ricci calculus | Matrix notation | Ricci calculus |
|---|---|---|---|
| $c = x^\top y$ | $c = x_i y^i$ | $A = xy^\top$ | $A_j^i = x^i y_j$ |
| $x = Ay$ | $x^i = A_j^i y^j$ | $z = x \odot y$ | $z^i = x^i y^i$ |
| $x^\top = y^\top A$ | $x_j = y_i A_j^i$ | $B = A\,\mathrm{diag}(x)$ | $B_j^i = A_j^i x_j$ |
| $C = A \cdot B$ | $C_k^i = A_j^i B_k^j$ | $B = \mathrm{diag}(x)A$ | $B_j^i = x^i A_j^i$ |

**Elements of Ricci calculus.** Only objects with exactly the same indices can be added or subtracted. For instance, the addition $x^i + y^i$ is a valid expression in Ricci calculus, but the expression $x^i + y^j$ is not. Obviously, also $x^i + x_i$ is not a valid expression, because there is no addition of vectors and covectors. The standard multiplication of objects that do not share an index is the outer product (or standard tensor product). For example, the product $x^i y_j$ describes a matrix that encodes a linear map. A multiplication that contains the same index once as an upper and once as a lower index describes an inner product, or contraction. A contraction entails the sum over all entries in the object that are addressed by the shared index. For instance, $x^i y_i$ encodes the evaluation of the linear form $y_i$ at the vector $x_i$, $A_i^j x^i$ encodes the evaluation of the linear map $A_i^j$ at the vector $x^i$. Note that, in contrast to standard matrix multiplication, multiplication in Ricci calculus is commutative. This will be very helpful later when we are deriving algorithms for computing derivatives. Applying element-wise unary functions like $\exp$ to some object keeps the indices, i.e., $\exp(A_j^i) = \exp(A)_j^i$.

Our version of Ricci calculus also features some special symbols. For instance, $0_j^i$ encodes the linear map that maps any vector to $0$. Hence, all entries of the matrix $0_j^i$ are $0$. Similarly, $0^i$ encodes the $0$-vector. The determinant of a matrix $A_i^j$ is denoted as $\det(A_i^j)$. The trace of $A_i^j$ does not need a special symbol in Ricci calculus, because it can be encoded by the expression $A_i^j \delta_j^i$.

**Extension to higher order tensors.** Since Ricci calculus can be extended easily to include more general tensor expressions we can also generalize our approach to computing gradients of more general tensor expressions. Higher order tensors just have more (free) indices. Note that for encoding higher order tensors, the use of some form of indices is unavoidable. Hence, we can use the Ricci calculus language directly for specifying such expressions and their derivatives; there is no need for translating them into another language. However, the translation makes sense for matrix expressions since the succinct, index free, standard language is very popular and heavily used in machine learning.

## 3 Tensor calculus

To process mathematical expressions, we usually represent them by a tree, or more generally, by a directed acyclic graph (DAG). The roots of the DAG, referred to as input nodes, have no parents and represent the variables. The leaves of the DAG, or output nodes, have no children and represent the functions that the DAG computes. Let the DAG have $n$ input nodes (variables) and $m$ output nodes (functions). We label the input nodes $x[0], ..., x[n-1]$, the output nodes $y[0], ..., y[m-1]$, and the internal nodes $v[0], ..., v[k-1]$. Every internal and every output node represents either a unary or a binary operator. The arguments of these operators are supplied by their parent nodes. Hence, every node in the DAG can have at most two parents, but multiple children. Every edge that connects a parent $x$ with a child $f$ is labeled by the easy to compute derivative of $f$ with respect to $x$, i.e., the derivative $\frac{\partial f}{\partial x}$. For instance, if $f = x \cdot y$ (multiplication operator), then the label of the edge $(x, f)$ is $y$. In case of $f = x + y$ (addition operator), the label of this edge would be $1$, and for $f = \sin(x)$ (sine operator), the label is $\cos(x)$. Figure 1 illustrates the case $f = x^\top A x$.

In automatic differentiation, one distinguishes between forward and reverse mode. Both modes are derived from the chain rule. In forward mode, the derivative of the roots is computed from roots to leaves along the edges and in reverse mode they are computed from leaves to roots. The edge labels are then multiplied along all paths and the products are summed up and stored in the nodes of the DAG.

**Forward mode.** In forward mode for computing derivatives with respect to the input variable $x[j]$, each node $v[i]$ will eventually store the derivative $\dot{v}[i] = \frac{\partial v[i]}{\partial x[j]}$, that is computed from root to leaves: At the root nodes representing the variables $x[i]$, the derivatives $\frac{\partial x[i]}{\partial x[j]}$ are stored. Then the derivatives that are stored at the remaining nodes, here called $f$, are iteratively computed by summing over all their incoming edges using the following equation:

$$\dot{f} = \frac{\partial f}{\partial x[j]} = \sum_{x \,:\, x \text{ is parent of } f} \frac{\partial f}{\partial x} \cdot \frac{\partial x}{\partial x[j]} = \sum_{x \,:\, x \text{ is parent of } f} \frac{\partial f}{\partial x} \dot{x}, \tag{1}$$

where the $\frac{\partial f}{\partial x}$ are the labels of the incoming edges of $f$ and the $\dot{x}$ have been computed before and are stored at the parent nodes $x$ of $f$. This means, the derivative of each function is stored at the corresponding leaves $y[i]$ of the expression DAG. Obviously, the updates can be done simultaneously for one input variable $x[j]$ and all output nodes $y[i]$. Computing the derivatives with respect to all input variables requires $n$ such rounds.

**Reverse mode.** Reverse mode automatic differentiation proceeds similarly, but from leaf to roots. Each node $v[i]$ will eventually store the derivative $\bar{v}[i] = \frac{\partial y[j]}{\partial v[i]}$, where $y[j]$ is the function to be differentiated. This is done as follows: First, the derivatives $\frac{\partial y[j]}{\partial y[i]}$ are stored at the leaves of the DAG. Then the derivatives that are stored at the remaining nodes, here called $x$, are iteratively computed by summing over all their outgoing edges using the following equation:

$$\bar{x} = \frac{\partial y[j]}{\partial x} = \sum_{f \,:\, f \text{ is child of } x} \frac{\partial y[j]}{\partial f} \cdot \frac{\partial f}{\partial x} = \sum_{f \,:\, f \text{ is child of } x} \bar{f} \cdot \frac{\partial f}{\partial x}, \tag{2}$$

where the $\frac{\partial f}{\partial x}$ are the labels of the outgoing edges of $x$ and the $\bar{f}$ have been computed before and are stored at the children $f$ of $x$. This means, that the derivative of the function $y[j]$ with respect to all the variables $x[i]$ is stored at the corresponding roots of the expression DAG. Computing the derivatives for all the output functions requires $m$ such rounds.

**Tensor calculus.** Tensor calculus can be implemented straightforwardly using either the forward or reverse mode. The input nodes are now considered tensors (symbols) like $x^i$ or $A_i^i$. These symbols are combined by the inner and the output nodes of the expression DAG into more complex tensor expressions like $A_i^j x^i$. In contrast to existing approaches, the output nodes can represent non-scalar tensor expressions, too. To compute the derivatives, the edges of the expression DAG are labeled by the corresponding tensor derivatives $\dot{v}[i]$ in forward mode and $\bar{v}[i]$ in reverse mode, respectively. Derivatives are then computed at expression level as described above. To the best of our knowledge, this is the first approach for applying automatic differentiation to matrix and tensor expressions that treats forward and reverse mode equally. We show in Section 4 that the symbolic expressions for higher order derivatives, obtained through this approach, can be evaluated very efficiently.

**Why is standard matrix calculus more complicated?** The standard language for matrix expressions in linear algebra is, as discussed in Section 2, not precise. The multiplication operator, in particular, is overloaded and does not refer to one but to several different multiplications. The following examples illustrate how this lack of precision complicates the process of computing derivatives.

Consider the simple matrix multiplication $C = AB$ as part of an expression DAG. It holds, that $\dot{C} = \dot{A}B + A\dot{B}$ for the forward mode and $\bar{A} = \bar{C}B^\top$ and $\bar{B} = A^\top \bar{C}$ for the reverse mode, see [8]. These equations cannot be instantiations of Equations (1) and (2) for the following reasons: Consider the two edges from the (parent) nodes that represent $A$ and $B$, respectively, to the node that represents $C = AB$. We have $\frac{\partial C}{\partial A} = B$ and $\frac{\partial C}{\partial B} = A$. In forward mode we have to multiply the differential $\dot{A}$ with $\frac{\partial C}{\partial A}$ from the right and the differential $\dot{B}$ with $\frac{\partial C}{\partial B}$ from the left. In Equation (1), though, both multiplications are always from the left. Similarly, in reverse mode, we also multiply once from the left and once from the right, while both multiplications are always from the right in Equation (2). Furthermore, the multiplication in reverse mode is not with $\frac{\partial C}{\partial A}$ and $\frac{\partial C}{\partial B}$, respectively, but with their transposes. This might seem negligible at first, to be fixed by slight adjustments to Equations (1) and (2). The expression $c = \det(A)$ shows that this is not so easy. The DAG for this expression has only two nodes, an input node (parent) that represents the matrix $A$ and its child (output node) that represents $c = \det(A)$. In forward mode, conventional approaches yield $\dot{c} = c\,\mathrm{tr}(\mathrm{inv}(A)\dot{A})$, yet $\bar{A} = \bar{c}\,c\,\mathrm{inv}(A)^\top$ in reverse mode, see again [8]. It is impossible to bring these equations into

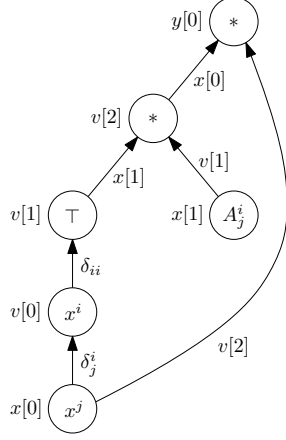

Figure 1: Expression DAG for $x^\top A x$ used for computing first order derivatives.

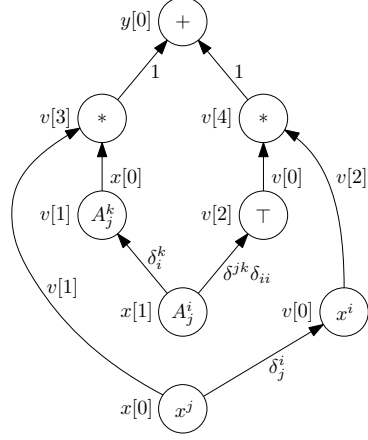

Figure 2: Expression DAG for $Ax + A^\top x$ used for computing second order derivatives.

the form of Equations (1) and (2). Using Ricci calculus, Equations (1) and (2) hold with the same edge label $\frac{\partial c}{\partial A_j^i} = \mathrm{adj}(A_j^i)$ for both, forward and reverse mode, where $\mathrm{adj}$ is the adjoint of the matrix $A_j^i$. We also want to point out that the equations from [8] that we used here in conjunction with standard matrix language are only valid for scalar expressions, i.e., scalar output functions. Our Ricci calculus-based approach accommodates matrix- or even higher order tensor-valued expressions and consequently higher order derivatives just as well. Hence, while standard matrix language is the preferred notation in linear algebra, it is not well suited for computing derivatives. Switching to Ricci calculus leads to a clean and elegant tensor calculus that is, without exceptions, rooted in Equations (1) and (2).

**Example.** To illustrate the tensor calculus algorithm we demonstrate it on the rather simple example $f = (x^\top A)x$ and compute derivatives of $f$ with respect to $x$. We provide an outline of the steps of the algorithm in computing the first and the second order derivatives through forward mode automatic differentiation. The individual steps of the reverse mode for this example can be found in the supplemental material.

First, the standard matrix language expression $(x^\top A)x$ is translated into its corresponding Ricci calculus expression $x_i A_j^i x^j$. Figure 1 shows the corresponding expression DAG and Table 2 shows the individual steps for computing the gradient with respect to $x^k$. The derivative can be read off from the last line of the table. Taking the transpose gives the gradient, i.e., $A_j^i x^j \delta_{ik} \delta^{kk} + x_i A_k^i \delta^{kk} = A_j^k x^j + A_k^i \delta^{kk} \delta_{ii} x^i$. Translating this expression back into matrix notation yields $Ax + A^\top x$.

Table 2: Individual steps of the forward mode automatic differentiation for $x^\top A x$ with respect to $x$.

| Forward trace | | | Forward derivative trace | | |
|---|---|---|---|---|---|
| $x[0] = x^j$ | | | $\dot{x}[0] = \delta_k^j$ | | |
| $x[1] = A_j^i$ | | | $\dot{x}[1] = 0_{jk}^i$ | | |
| $v[0] = x[0]\delta_j^i$ | $= x^i$ | | $\dot{v}[0] = \dot{x}[0]\delta_j^i$ | $= \delta_k^i$ | |
| $v[1] = v[0]\delta_{ii}$ | $= x_i$ | | $\dot{v}[1] = \dot{v}[0]\delta_{ii}$ | $= \delta_{ik}$ | |
| $v[2] = v[1]x[1]$ | $= x_i A_j^i$ | | $\dot{v}[2] = \dot{v}[1]x[1] + \dot{x}[1]v[1]$ | $= A_j^i \delta_{ik} + 0_{jk}$ | |
| $y[0] = v[2]x[0]$ | $= x_i A_j^i x^j$ | | $\dot{y}[0] = \dot{v}[2]x[0] + \dot{x}[0]v[2]$ | $= A_j^i x^j \delta_{ik} + x_i A_k^i$ | |

To obtain the Hessian we just compute the derivative of $A_j^k x^j + A_k^i \delta^{kk} \delta_{ii} x^i$ with respect to $x^l$. Figure 2 shows the corresponding expression DAG and Table 3 contains the individual steps for computing the Hessian that can be read off the last line of the table as $A_l^k + (A_k^l)^\top$. Translating back to matrix notation and taking the transpose yields the desired result $A + A^\top$. Note that we simplified the expression in Table 3 by writing $(A_k^i)^\top$ instead of $A_k^i \delta^{kk} \delta_{ii}$.

Table 3: Individual steps of the forward mode computation of the Hessian of $x^\top A x$ with respect to $x$.

| Forward trace | | Forward derivative trace | |
|---|---|---|---|
| $x[0] = x^j$ | | $\dot{x}[0] = \delta^j_l$ | |
| $x[1] = A^i_j$ | | $\dot{x}[1] = 0^i_{jl}$ | |
| $v[0] = x[0]\delta^i_j$ | $= x^i$ | $\dot{v}[0] = \dot{x}[0]\delta^i_j$ | $= \delta^i_l$ |
| $v[1] = x[1]\delta^k_i$ | $= A^k_j$ | $\dot{v}[1] = \dot{x}[1]\delta^k_i$ | $= 0^k_{jl}$ |
| $v[2] = x[1]\delta^{jk}\delta_{ii}$ | $= (A^i_k)^\top$ | $\dot{v}[2] = \dot{x}[1]\delta^{jk}\delta_{ii}$ | $= 0^k_{il}$ |
| $v[3] = x[0]v[1]$ | $= A^k_j x^j$ | $\dot{v}[3] = \dot{x}[0]v[1] + \dot{v}[1]x[0]$ | $= A^k_l + 0^k_l$ |
| $v[4] = v[2]v[0]$ | $= (A^i_k)^\top x^i$ | $\dot{v}[4] = \dot{v}[2]v[0] + \dot{v}[0]v[2]$ | $= 0^k_l + (A^l_k)^\top$ |
| $y[0] = v[3] + v[4]$ | $= A^k_j x^j + (A^i_k)^\top x^i$ | $\dot{y}[0] = 1 \cdot \dot{v}[3] + 1 \cdot \dot{v}[4]$ | $= A^k_l + (A^l_k)^\top$ |

# 4 Experiments

We have implemented our algorithms in Python. To evaluate expression graphs we use the NumPy and CuPy packages. Our framework can perform forward mode as well as reverse mode automatic differentiation. Reverse mode is the more involved mode (it has a forward and a backward pass) and it is also the one that is commonly used since it allows to compute derivatives with respect to many input variables simultaneously. Hence, in the experiments we only use reverse mode automatic differentiation. Similarly to all other frameworks, our implementation also performs some expression simplifications, i.e., constant folding, pruning of zero tensors, and removal of multiplication by $\delta$ tensors when applicable. An interface to our framework for computing vector and matrix derivatives is available online at www.MatrixCalculus.org.

**Experimental set up.** We compare our framework to the state-of-the-art automatic differentiation frameworks TensorFlow 1.10, PyTorch 0.4, Theano 1.0, and HIPS autograd 1.2 used with Python 3.6, that were all linked against Intel MKL. All these frameworks support reverse mode automatic differentiation for computing first order derivatives. They all compute the Hessian row by row by iteratively computing products of the Hessian with standard basis vectors. All frameworks provide interfaces for computing Hessians except PyTorch. Here, we follow the instructions of its developers. The experiments were run in a pure CPU setting (Intel Xeon E5-2686, four cores) as well as in a pure GPU setting (NVIDIA Tesla V100), except for autograd, that does not provide GPU support.

Hessians are not needed for large-scale problems that typically arise in deep learning. However, in optimization problems arising from 'classical' machine learning problems like logistic regression, elastic net, or inference in graphical models, optimization algorithms based on Newton steps can be faster than gradient based algorithms when the number of optimization variables is in the few thousands. The Newton steps entail solving a system of linear equations. A direct solver for such systems of moderate dimension can be much faster than an iterative solver that computes Hessian-vector products. This is particularly true for ill-conditioned problems. Hence, we chose three representative 'classical' machine learning problems for our experiments, namely quadratic functions, logistic regression, and matrix factorization. For these problems we ran two sets of experiments. In the first set we measured the running times for evaluating the function value and the gradient together, and in a second set we measured the running times for evaluating the Hessian. The evaluation of Jacobians is implicitly covered in the experiments since all frameworks run the same code both for evaluating Jacobians and for evaluating Hessians. Hence, we restrict ourselves to Hessians here.

The measurements do not include the time needed for constructing the expression graphs for the gradients and the Hessians. TensorFlow and Theano took a few seconds to create the expression graphs for the second order derivatives while our approach only needed roughly 20 milliseconds. PyTorch and HIPS autograd create the expression graph for the derivative dynamically while evaluating the function, hence, its creation time cannot be determined.

**Quadratic function.** The probably most simple function that has a non-vanishing Hessian is the quadratic function $x^\top A x$ that we have been using as an example throughout this paper. Several classical examples from machine learning entail optimizing a quadratic function, for instance, the dual of a support vector machine [5], least squares regression, LASSO [24], or Gaussian processes [19], to name just a few. Of course, the Hessian of a quadratic function can be easily computed by

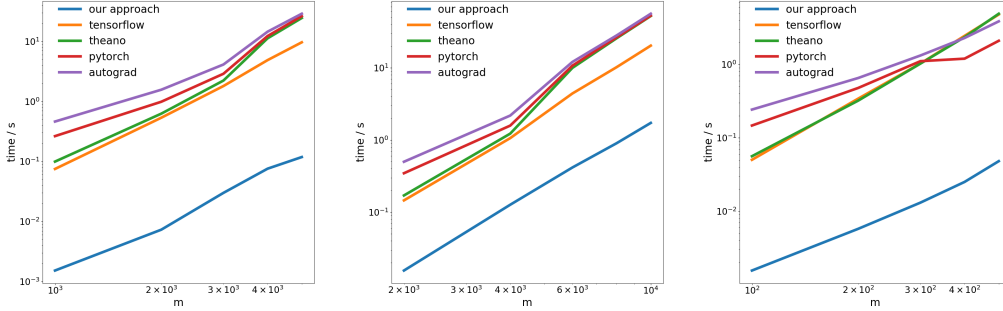

Figure 3: Log-log plot of the running times for evaluating the Hessian of the quadratic function (left), logistic regression (middle), matrix factorization (right) on the CPU. See the supplemental material for a table with the running times.

hand. However, here we want to illustrate that even for this simple example, running times can vary dramatically.

**Logistic regression.** Logistic regression [6] is probably one of the most commonly used methods for classification. Given a set of $m$ data points $X \in \mathbb{R}^{m \times n}$ along with a set of binary labels $y \in \{\pm 1\}^m$, logistic regression aims at minimizing the loss function $\sum_i \log \left( \exp \left( -y^{(i)} \left( X^{(i)} w \right) \right) + 1 \right)$, where $w \in \mathbb{R}^n$ is the weight vector, $X^{(i)}$ is the $i$-th data point ($i$-th row of $X$), and $y^{(i)}$ the corresponding $i$-th label. The data matrix $X$ can be composed of the true input features, features transformed by basis functions/kernels [4, 21], or by random basis functions [18], or by features that have been learned by a deep net [11]. We set $m = 2n$ in the experiments.

**Matrix factorization.** Matrix factorization can be stated as the problem $\min_{U,V} \|T - UV^\top\|_\Omega^2$, where $T \in \mathbb{R}^{m \times n}$ is some target matrix, $U \in \mathbb{R}^{m \times k}$ and $V \in \mathbb{R}^{n \times k}$ are the low-rank factor matrices, and $\Omega \in \{0,1\}^{m \times n}$ is an indicator matrix that defines which elements of $T$ are known. Matrix factorization is mainly used in the context of recommender systems [13] or natural language processing [3, 12]. For the experiments, we set $k = 5$ and compute the gradient and Hessian with respect to $U$. Note that the Hessian is a fourth order tensor. In Ricci calculus it reads as $2\delta_m^n \delta_l^j V_i^k V_i^j \delta_{jj} \delta^{ii}$, or in our slightly abbreviated notation as $2\delta_m^n \delta_l^j V_i^k (V_i^j)^\top$.

**Results.** The experiments show that basically all frameworks are equally fast when evaluating first order derivatives. This is no longer true in the case of second order derivatives like Hessians. Since our approach extends naturally to higher order derivatives, it should not surprise that it is faster than the reference frameworks. Indeed, as can be seen in Figure 3, it is up to two orders of magnitude faster than the existing frameworks on the CPU. On the GPU, the speedup is about three orders of magnitude, see the supplemental material, where you also find the remaining results and more details. The reason for this speedup is the fact, that our approach is able to produce compact matrix expressions that can be mapped to efficient BLAS implementations whereas all other approaches produce fairly large expression graphs whose evaluation involves many for-loops and complex memory access. The GPU can deal with this even worse which leads to even larger speed-ups on the GPU.

## 5 Conclusion

We have presented the first algorithmic framework for computing matrix and tensor derivatives that naturally extends to higher order derivatives. Experiments show that our approach achieves state-of-the-art performance for the evaluation of first order derivatives. In the case of second order derivatives it is up to two orders of magnitude more efficient on CPUs and up to three orders of magnitude more efficient on GPUs.

Our framework operates directly on tensors using Ricci calculus to specify tensor expressions. Operating on tensors also enables classical, symbolic matrix calculus that appears notoriously difficult in standard matrix language. We showed that the difficulties in computing matrix derivatives can be attributed to the predominantly used standard matrix language that, in contrast to Ricci calculus, is not well suited for matrix calculus.

## Acknowledgments

Sören Laue has been funded by Deutsche Forschungsgemeinschaft (DFG) under grant LA 2971/1-1. This work has also been supported by the AWS Cloud Credits for Research program and by a gift from Google.

## Footnotes

[1]Earlier versions of SymPy contained an algorithm for computing matrix derivatives. However, since it sometimes produced erroneous results it has been removed, see, e.g., https://github.com/sympy/sympy/issues/10509.

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
