[Supplementary Material · supplement.pdf]

# Computing Higher Order Derivatives of Matrix and Tensor Expressions (supplemental material)

## Section 3: Tensor calculus (example reverse mode)

In the paper we have demonstrated *forward* mode automatic differentiation on the example of computing first and second order derivatives of the expression $f = (x^\top A)x$ with respect to $x$, see Figure 1 in the paper for the corresponding expression DAG. Here, we show all the steps for computing first and second order derivatives of $f$ by *reverse* mode automatic differentiation with respect to $x$ and also with respect to $A$. Table 1 shows the individual steps for computing the gradients and Table 2 shows the individual steps for computing second order derivatives.

**First order derivatives.** From Table 1 we can read off the derivative with respect to $x$ and also with respect to $A$. The table states that $\bar{x}[0] = x^j A^i_j \delta_{ik} + x_i A^i_k$. Taking the transpose of this expression yields the gradient with respect to $x$, i.e., $A^i_j x^j \delta_{ik} \delta^{kk} + x_i A^i_k \delta^{kk} = A^i_j x^j + A^i_k \delta^{kk} \delta_{ii} x^i$, or in matrix notation $Ax + A^\top x$. We get the derivative with respect to $A$ from $\bar{x}[1] = x^l x_k$. Taking the transpose of this expression and transforming it into matrix notation yields $xx^\top$ for the derivative with respect to $A$.

Table 1: Individual steps of the reverse mode automatic differentiation for $x^\top A x$.

| Forward trace | | Reverse trace | |
|---|---|---|---|
| $x[0] = x^j$ | | $\bar{x}[0] = \bar{v}[0]\delta^i_j \delta^j_k + \bar{y}[0]v[2]\delta^j_k$ | $= x^j A^i_j \delta_{ik} + x_i A^i_k$ |
| $x[1] = A^i_j$ | | $\bar{x}[1] = \bar{v}[2]v[1]\delta^i_k \delta^l_j$ | $= x^l x_k$ |
| $v[0] = x[0]\delta^i_j$ | $= x^i$ | $\bar{v}[0] = \bar{v}[1]\delta_{ii}$ | $= x^j A^i_j \delta_{ii}$ |
| $v[1] = v[0]\delta_{ii}$ | $= x_i$ | $\bar{v}[1] = \bar{v}[2]x[1]$ | $= x^j A^i_j$ |
| $v[2] = v[1]x[1]$ | $= x_i A^i_j$ | $\bar{v}[2] = \bar{y}[0]x[0]$ | $= x^j$ |
| $y[0] = v[2]x[0]$ | $= x_i A^i_j x^j$ | $\bar{y}[0] = 1$ | |

**Second order derivatives.** As examples for second order derivatives, we compute the derivative of $A^k_j x^j + A^i_k \delta^{kk} \delta_{ii} x^i$, see Figure 2 in the paper for the corresponding expression DAG, with respect to $x^l$ and also with respect to $A^l_m$. Table 2 shows the individual steps of the reverse mode, where we have replaced the expression $A^i_k \delta^{kk} \delta ii$ by $(A^i_k)^\top$ for a more succinct representation.

Note that the derivative with respect to $x^l$ yields the Hessian of $f$ with respect to $x$. From $\bar{x}[0]$ in the first row in the second column of Table 2 we can read off this Hessian as $A^k_l + (A^l_k)^\top$. Translating this expression back to matrix notation and taking the transpose yields the expected result $A + A^\top$.

The second order derivative $\frac{\partial^2 f}{\partial x_k \partial A^l_m}$ can be read off from $\bar{x}[1]$ in the second row in the second column of Table 2 as $\delta^k_l x^m + x_l \delta^{km}$. This derivative is a third order tensor that cannot be represented in standard matrix notation.

Table 2: Individual steps of the forward mode for the Hessian of $x^\top A x$ with respect to $x$.

| Forward trace | | Reverse trace | |
|---|---|---|---|
| $x[0] = x^j$ | | $\bar{x}[0] = \bar{v}[3]v[1]\delta_l^j + \bar{v}[0]\delta_j^i\delta_l^j$ | $= A_l^k + (A_k^l)^\top$ |
| $x[1] = A_j^i$ | | $\bar{x}[1] = \bar{v}[1]\delta_i^k\delta_l^i\delta_j^m + \bar{v}[2]\delta^{jk}\delta_{ii}\delta_l^i\delta_j^m$ | $= x^m\delta_l^k + x_l\delta^{mk}$ |
| $v[0] = x[0]\delta_j^i$ | $= x^i$ | $\bar{v}[0] = \bar{v}[4]v[2]$ | $= (A_k^i)^\top$ |
| $v[1] = x[1]\delta_i^k$ | $= A_j^k$ | $\bar{v}[1] = \bar{v}[3]x[0]$ | $= x^j$ |
| $v[2] = x[1]\delta^{jk}\delta_{ii}$ | $= (A_k^i)^\top$ | $\bar{v}[2] = \bar{v}[4]v[0]$ | $= x^i$ |
| $v[3] = x[0]v[1]$ | $= A_j^k x^j$ | $\bar{v}[3] = \bar{y}[0]\cdot 1$ | $= 1$ |
| $v[4] = v[2]v[0]$ | $= (A_k^i)^\top x^i$ | $\bar{v}[4] = \bar{y}[0]\cdot 1$ | $= 1$ |
| $y[0] = v[3] + v[4]$ | $= A_j^k x^j + (A_k^i)^\top x^i$ | $\bar{y}[0] = 1$ | |

# Section 4: Experiments

Here we provide the missing results for the first set of experiments, where we measured the time for evaluating function values and gradients for the example problems (quadratic function, logistic regression, and matrix factorization). We also report the results for the second set of experiments, namely the running times for evaluating Hessians on the GPU.

For the general set up of the experiments please refer to the paper. Here we only want to add that in order to account for outliers and noise, the reported running times were obtained as follows: Each experiment was run in ten batches of five runs each. The running times for each batch were averaged, and the three batches with the worst average running times were not considered. Hence, the reported running times are the average of the average running times of the seven best batches. Such a setup also removes the overhead that is caused by TensorFlow of optimizing the execution of the expression graph on the GPU. The first few evaluations of the expression graph can be slower than consecutive ones because TensorFlow optimizes the execution schedule based on the first runs. Hence, we automatically take only the optimized runs into account.

**First set of experiments.** The measured running times for evaluating function values and gradients on the CPU are shown in Figure 1 and reported in Table 3, and the corresponding running times on the GPU are shown in Figure 2 and reported in Table 4.

**Second set of experiments.** The measured running times for evaluating Hessians on the CPU are shown in Figure 3 (in the paper) and reported in Table 5, and the corresponding running times on the GPU are shown in Figure 3 and reported Table 6.

**Discussion.** As we have already discussed in the paper, in general we observe that all frameworks perform roughly similarly when evaluating function values and gradients on the CPU, see Figure 1. For the quadratic function and the logistic function the difference in speed between the fastest and the slowest framework is about a factor of two. For the matrix factorization problem our approach is roughly three times faster than the best competing framework. Still, we would call all frameworks equally fast. On the GPU all frameworks also perform similarly good when computing function values and gradients as can be observed in Figure 2. Our framework is now about a factor of two slower than the fastest framework. This is not surprising since the GPU support of our framework is simply through CuPy without any tuning. However, in this paper we focus on the algorithmic improvement and not on an engineering improvement.

The situation changes when we consider higher order derivatives, here more specifically the evaluation of Hessians. Our approach is up to two orders of magnitude more efficient than the existing frameworks for evaluating Hessians on the CPU. The advantage of our approach is even more pronounced on the GPU, where we observe a speed up of up to three orders of magnitude. As we have pointed out in the paper the reason for this comes from the fact, that our approach is able to produce compact matrix expressions that can be mapped to efficient BLAS implementations as opposed to large expression graphs.

# Figures for function value and gradient experiments on the CPU and GPU

Figure 1: Log-log plot of the running times for evaluating the function value and gradient on the CPU for the quadratic function (left), logistic regression (middle), and matrix factorization (right).

Figure 2: Log-log plot of the running times on the GPU for evaluating the function value and gradient for the quadratic function (left), logistic regression (middle), and matrix factorization (right)

# Figures for Hessian experiments on the GPU

Figure 3: Log-log plot of the running times on the GPU for evaluating the Hessian for the quadratic function (left), logistic regression (middle), and matrix factorization (right).

# Tables for function value and gradient experiments on the CPU

Table 3: Running times for evaluating function value and gradient on the CPU

Quadratic function with $x \in \mathbb{R}^m$ and $A \in \mathbb{R}^{m \times m}$

| $m$ | our approach | TensorFlow | Theano | PyTorch | autograd |
|---|---|---|---|---|---|
| 10,000 | 0.020 | 0.034 | 0.030 | 0.021 | 0.055 |
| 15,000 | 0.047 | 0.081 | 0.070 | 0.064 | 0.075 |
| 20,000 | 0.084 | 0.150 | 0.124 | 0.150 | 0.127 |
| 25,000 | 0.129 | 0.228 | 0.192 | 0.236 | 0.193 |
| 30,000 | 0.189 | 0.344 | 0.285 | 0.349 | 0.287 |

Logistic regression with $X \in \mathbb{R}^{m \times n}, w \in \mathbb{R}^n$, and $y \in \{\pm 1\}^m$

| $m$ | $n$ | our approach | TensorFlow | Theano | PyTorch | autograd |
|---|---|---|---|---|---|---|
| 10,000 | 5,000 | 0.0097 | 0.0199 | 0.0152 | 0.0109 | 0.0152 |
| 12,000 | 6,000 | 0.0142 | 0.0328 | 0.0217 | 0.0151 | 0.0219 |
| 14,000 | 7,000 | 0.0196 | 0.0421 | 0.0300 | 0.0213 | 0.0298 |
| 16,000 | 8,000 | 0.0261 | 0.0467 | 0.0390 | 0.0274 | 0.0391 |
| 18,000 | 9,000 | 0.0332 | 0.0613 | 0.0499 | 0.0305 | 0.0497 |
| 20,000 | 10,000 | 0.0408 | 0.0790 | 0.0605 | 0.0474 | 0.0607 |
| 22,000 | 11,000 | 0.0498 | 0.0851 | 0.0750 | 0.0511 | 0.0738 |
| 24,000 | 12,000 | 0.0597 | 0.1009 | 0.0914 | 0.0612 | 0.0882 |

Matrix factorization with $U \in \mathbb{R}^{m \times k}, V \in \mathbb{R}^{n \times k}, T \in \mathbb{R}^{m \times n}$, and $k = 5$

| $m = n$ | our approach | TensorFlow | Theano | PyTorch | autograd |
|---|---|---|---|---|---|
| 5,000 | 0.050 | 0.209 | 0.402 | 0.140 | 0.150 |
| 6,000 | 0.074 | 0.301 | 0.582 | 0.168 | 0.216 |
| 7,000 | 0.101 | 0.408 | 0.791 | 0.222 | 0.292 |
| 8,000 | 0.132 | 0.532 | 1.035 | 0.364 | 0.382 |
| 9,000 | 0.167 | 0.673 | 1.310 | 0.474 | 0.481 |
| 10,000 | 0.208 | 0.830 | 1.649 | 0.573 | 0.594 |

# Tables for function value and gradient experiments on the GPU

Table 4: Running times for evaluating function value and gradient on the GPU

Quadratic function with $x \in \mathbb{R}^m$ and $A \in \mathbb{R}^{m \times m}$

| $m$ | our approach | TensorFlow | Theano | PyTorch |
|---|---|---|---|---|
| 10,000 | 0.0016 | 0.0015 | 0.0015 | 0.0013 |
| 20,000 | 0.0087 | 0.0043 | 0.0056 | 0.0038 |
| 30,000 | 0.0200 | 0.0096 | 0.0132 | 0.0086 |
| 40,000 | 0.0346 | 0.0157 | 0.0224 | 0.0152 |

Logistic regression with $X \in \mathbb{R}^{m \times n}, w \in \mathbb{R}^n$, and $y \in \{\pm 1\}^m$

| $m$ | $n$ | our approach | TensorFlow | Theano | PyTorch |
|---|---|---|---|---|---|
| 10,000 | 5,000 | 0.0010 | 0.0010 | 0.0008 | 0.0009 |
| 20,000 | 10,000 | 0.0049 | 0.0024 | 0.0028 | 0.0021 |
| 30,000 | 15,000 | 0.0112 | 0.0052 | 0.0068 | 0.0047 |
| 40,000 | 20,000 | 0.0272 | 0.0084 | 0.0112 | 0.0076 |

Matrix factorization with $U \in \mathbb{R}^{m \times k}, V \in \mathbb{R}^{n \times k}, T \in \mathbb{R}^{m \times n}$, and $k = 5$

| $m = n$ | our approach | TensorFlow | Theano | PyTorch |
|---|---|---|---|---|
| 6,000 | 0.00037 | 0.0035 | 0.00018 | 0.0026 |
| 8,000 | 0.00037 | 0.0063 | 0.00017 | 0.0045 |
| 10,000 | 0.00038 | 0.0092 | 0.00017 | 0.0089 |
| 12,000 | 0.00038 | 0.0129 | 0.00017 | 0.0123 |

# Tables for Hessian experiments on the CPU

Table 5: Running times for evaluating Hessians on the CPU

Quadratic function with $x \in \mathbb{R}^m$ and $A \in \mathbb{R}^{m \times m}$

| $m$ | our approach | TensorFlow | Theano | PyTorch | autograd |
|---|---|---|---|---|---|
| 1,000 | 0.0015 | 0.0750 | 0.0995 | 0.2644 | 0.4626 |
| 2,000 | 0.0073 | 0.5356 | 0.6261 | 0.9883 | 1.5680 |
| 3,000 | 0.0299 | 1.8064 | 2.2292 | 2.8921 | 4.1246 |
| 4,000 | 0.0758 | 4.8774 | 11.260 | 12.283 | 14.510 |
| 5,000 | 0.1184 | 9.6634 | 24.555 | 26.322 | 28.919 |

Logistic regression with $X \in \mathbb{R}^{m \times n}, w \in \mathbb{R}^n$, and $y \in \{\pm 1\}^m$

| $m$ | $n$ | our approach | TensorFlow | Theano | PyTorch | autograd |
|---|---|---|---|---|---|---|
| 2,000 | 1,000 | 0.015 | 0.145 | 0.170 | 0.346 | 0.500 |
| 4,000 | 2,000 | 0.127 | 1.063 | 1.227 | 1.585 | 2.183 |
| 6,000 | 3,000 | 0.416 | 4.417 | 9.996 | 10.72 | 12.07 |
| 8,000 | 4,000 | 0.902 | 10.20 | 25.67 | 26.38 | 28.06 |
| 10,000 | 5,000 | 1.727 | 20.31 | 52.26 | 52.89 | 56.20 |

Matrix factorization with $U \in \mathbb{R}^{m \times k}, V \in \mathbb{R}^{n \times k}, T \in \mathbb{R}^{m \times n}$, and $k = 5$

| $m = n$ | our approach | TensorFlow | Theano | PyTorch | autograd |
|---|---|---|---|---|---|
| 100 | 0.0015 | 0.0501 | 0.0561 | 0.1470 | 0.2426 |
| 200 | 0.0057 | 0.3425 | 0.3217 | 0.4787 | 0.6541 |
| 300 | 0.0130 | 1.0306 | 1.0273 | 1.1085 | 1.3250 |
| 400 | 0.0250 | 2.4542 | 2.3589 | 1.1996 | 2.2952 |
| 500 | 0.0480 | 4.7998 | 4.9174 | 2.1030 | 3.8720 |

## Tables for Hessian experiments on the GPU

Table 6: Running times for evaluating Hessians on the GPU

Quadratic function with $x \in \mathbb{R}^m$ and $A \in \mathbb{R}^{m \times m}$

| $m$ | our approach | TensorFlow | Theano | PyTorch |
|---|---|---|---|---|
| 2,000 | 0.0001 | 0.1620 | 0.1836 | 0.6968 |
| 4,000 | 0.0003 | 0.7360 | 0.8391 | 1.6399 |
| 6,000 | 0.0008 | 2.2008 | 2.3612 | 4.4334 |
| 8,000 | 0.0017 | 4.9623 | 5.1882 | 7.5246 |
| 10,000 | 0.0029 | 9.4902 | 9.8004 | 13.584 |

Logistic regression with $X \in \mathbb{R}^{m \times n}, w \in \mathbb{R}^n$, and $y \in \{\pm 1\}^m$

| $m$ | $n$ | our approach | TensorFlow | Theano | PyTorch |
|---|---|---|---|---|---|
| 4,000 | 2,000 | 0.0026 | 0.2645 | 0.2889 | 0.8191 |
| 8,000 | 4,000 | 0.0202 | 1.3833 | 1.4549 | 2.3199 |
| 12,000 | 6,000 | 0.0611 | 4.4498 | 4.5821 | 10.3117 |
| 16,000 | 8,000 | 0.1389 | 9.7879 | 9.9819 | 11.2283 |
| 20,000 | 10,000 | 0.2760 | 18.8960 | 19.1196 | 20.7802 |

Matrix factorization with $U \in \mathbb{R}^{m \times k}, V \in \mathbb{R}^{n \times k}, T \in \mathbb{R}^{m \times n}$, and $k = 5$

| $m = n$ | our approach | TensorFlow | Theano | PyTorch |
|---|---|---|---|---|
| 200 | 0.0005 | 0.2079 | 0.1695 | 0.6208 |
| 400 | 0.0007 | 0.4093 | 0.3750 | 1.1400 |
| 600 | 0.0007 | 0.6544 | 0.6645 | 1.7858 |
| 800 | 0.0008 | 0.8818 | 1.0033 | 2.3074 |
| 1000 | 0.0012 | 1.1191 | 1.4180 | 3.1574 |