[Reviews · NeurIPS 2018]

Reviewer 1



This paper introduces a framework for efficient matrix and tensor differentiation. The main conceptual contribution, in comparison to existing automatic differentiation frameworks, is to work with expressions in Ricci calculus, which explicitly distinguish between covariant and contravariant indices. As tensor contraction is associative and commutative, this results in an elegant, expressive, and principled way to do automatic differentiation on tensor expressions, compatible with forward-mode, backward-mode, and symbolic differentiation. I believe this work is a useful and exciting contribution to the ML community at large. The authors have clearly put thoughtful and extensive engineering effort into this work, and go as far as to provide an anonymized web API for their implementation of symbolic differentiation using this framework. However, I am somewhat concerned about its suitability with regard to the scope of ICML. Software framework papers (including related works cited by this paper) have tended to appear in other conferences. Though the work is clearly applicable in ML systems, its originality seems to derive chiefly from the interface design. Some comments, orthogonal to the above: - Since I think this falls in the category of a "systems paper", a more in-depth discussion of performance concerns might be warranted. In particular, it would be enlightening to see where the performance boost over existing frameworks comes from, especially in the GPU experiments in the appendix. If the time bottleneck consists of CUDA linear algebra operations, then does this framework produce expressions that are less redundant to evaluate? The paper's conceptual advantages would be greatly highlighted by a concrete analysis of how each other framework is handling Hessians suboptimally. On a similar note, a more thorough documentation of the simplifications and optimizations would be helpful. - Another way to make the paper a better fit for the ICML scope might be to outline an end-to-end optimization use case for the framework, showing a performance boost on some end-to-end task. - Though full second-order methods are currently non-standard in deep learning due to high model dimension, there has been some attention paid to optimization algorithms based on the primitives of Hessian-vector products (see, e.g., [1]). It would be timely and exciting to see Hessian-vector products implemented in this framework, and a performance comparison. As it stands, a state-of-the-art comparison on large-scale models is somewhat difficult, as is reflected by the instance sizes described in the appendix. - The paper is very clearly written, and provides a precise and helpful exposition of background knowledge. [1] N. Agarwal, Z. Allen-Zhu, B. Bullins, E. Hazan, T. Ma. Finding Approximate Local Minima Faster than Gradient Descent. STOC, 2017.

Reviewer 2



Edit: I have read the authors feedback. The authors propose a framework to compute high order derivative of function specified by a computation DAG. Their approach rely on two crucial insights: - This requires to be able to differentiate matrix expressions - Common matrix calculus language is not accurate enough to express this. The authors propose to use Ricci calculus, which was famously applied to general relativity and differential geometry. They describe an automatic differentiation framework based on this tool as well as simple differentiation rules. They illustrate the presented concepts based on simple examples and present numerical results suggesting that the proposed framework significantly outperforms existing architectures for automatic differentiation when applied to vector expressions (here a gradient to obtain a Hessian). Main comments: This paper was really convincing and pleasant to read. The context, concepts and problems are explained in a clear way. The proposed approach is illustrated through pedagogic examples. The numerical results speak for themselves. I clearly see the potential of the proposed framework to broaden and enhance accessibility to learning techniques through efficient and flexible computational tools. I think that this constitutes a very relevant contribution. On the other hand my knowledge about automatic differentiation is close to negligible so I cannot really question the novelty of the approach beyond the convincing arguments given in the paper. I am a bit surprised not to see referecences recently published in the litterature context paragraph. See details bellow. Furthermore, I am not competent to judge about potential loopholes in the numerical experiments, beyond, again, the fact that they are presented in a very convincing way. Minor questions: - The authors limit themselves to binary operators. What is the impact of this and how does it limit the approach. - Could the author explain why they only focus on backward differentiation? - The experiments do not account for the complexity of constructing the expression graph. Could the authors give a sense about how high the corresponding computational cost is and how it compares to concurent approaches? Biblio: A. Griewank and U. Naumann, Accumulating Jacobians as chained sparse matrix products, Mathematical Programming, 95 (2003), pp. 555– 571. R. M. Gower and M. P. Mello, A new framework for the computation of Hessians, Optimization Methods and Software, 27 (2012), pp. 251–273. Gower, R. M., & Gower, A. L. (2016). Higher-order reverse automatic differentiation with emphasis on the third-order. Mathematical Programming, 155(1-2), 81-103. Walther, A. (2008). Computing sparse Hessians with automatic differentiation. ACM Transactions on Mathematical Software (TOMS), 34(1), 3.

Reviewer 3



The paper presents an algorithm for computing higher order derivatives of multiple outputs, by translating matrix expressions into Ricci calculus, to generalize modern frameworks for automatic differentiation from first order derivatives of scalar outputs to higher order derivatives of multiple outputs. Overall comments ---------------- I really enjoyed the core idea of this paper; a generalized form of autograd has immense potential to increase accessibility to many higher order optimization methods by removing the implementation barrier, a benefit that many gradient-based optimization procedures enjoy today. The introduction to Ricci calculus was also easy to read, and the tensor calculus algorithm is easy to understand. However, I have a number of concerns on the submitted paper, largely regarding the experiments presented in the paper. 1) The paper describes a forward mode, but does not present any sort of evaluation thereof. Jacobians and higher order derivatives are mentioned in the introduction but the experiments focus solely on Hessian computation. The proposed method claims to be able compute derivatives of non-scalar valued functions, but the experiments only demonstrate the approach on scalar-valued functions. The paper would be much stronger if the empirical results matched the described abilities. 2) The main paper claims even better speed up on the GPU. Why do the authors instead present the supposedly worse CPU results up front? 3) I could not find many details regarding the experimental setup, e.g. where the data for the problem came from, whether they are real or synthetic, how the problems are generated if synthetic, etc. 4) There is some strange behavior in Figure 2 of the supplemental material. The authors write that the worse performance of Tensorflow and PyTorch is due to unavoidable data transfer from the main memory and the GPU. However, for some reason PyTorch doesn't suffer from this (but tensorflow does) in only the left-most plot. Why is this the case? Also, within these two frameworks, it should be possible to transfer all data to the GPU before timing the functions, and it is definitely possible to directly allocating all future tensors on the GPU, without needing intermediate data transfers. If the authors believe this is not possible, please elaborate why. 5) The authors point out that one of the drawbacks of existing tools is the creation of enormous computational graphs when computing Hessians. However, the authors explicitly do not report the time for constructing the expression graphs on line 287. For better empirical motivation, the paper should presents timing results that quantify and back up this statement. The results should also compare to the baseline approach of computing the terms based on a formula derived by hand to demonstrate how much (or little) overhead the proposed algorithm has. Overall, while I enjoyed the exposition on Ricci calculus and automatic differentiation, it is perhaps too verbose, since the paper leaves very little room for experimental justification. This results in a paper that, while well motivated, is experimentally weak. While the few experiments presented do show promise, I cannot recommend acceptance solely based on this, as a much more compelling case can likely be given. Hopefully the authors can shed some light in the rebuttal, as the work here seems to be an important next step in autodifferentiation tools for the machine learning community. Minor comments -------------- a) Can you elaborate more on the translation to and from Ricci calculus? To actually compute the terms, are expressions translated back into matrix algebra before being evaluated, or are you evaluating the Ricci calculus expressions directly? When you evaluate the Ricci expressions, how do you decide an order of evaluation? For tensor expressions, such choices can greatly effect the actual computational cost of evaluating the expression. b) Consider include benchmarking results on computing Hessian-vector products, which is also a common use-case when the explicit Hessian cannot be formed or directly decomposed (e.g. due to memory or computational constraints). c) In the web demo, it would be nice to include more objectives of the mentioned 'classical' machine learning problems (listed in lines 276-277) as examples, as well as examples which compute Hessians as opposed to gradients, and examples which compute non-scalar outputs. It also appears that there is no max operator. Quality ------- Overall the experiments section is much poorer in quality, as described in this review. Clarity ------- I found the majority of the main paper to be clear and well-written, with one major exception: the experiments section. Specifically, the `Results' paragraph is extremely poorly written, with most of the sentences not being well-formed. One other major point of clarity is that there is a mismatch between the statements in the conclusion and the experiments. The conclusion claims that the approach achieves state of the art performance for first order derivatives, while the experiments merely claim that 'basically, all frameworks are equally fast'. The plots in the Appendix do not back up the conclusion. minor suggestions for main paper: -On line 115, it may not be clear to readers what it means to 'transform contravariantly' or 'transform covariantly' -The last sentence in line 273-274 does not make sense. -While 'probably' and 'basically' are not necessarily bad to use in general, their usage in the experimental section is not great -Figure 3 needs to say how large n is, which I could only found in the Appendix. -The sentence ending on line 232 is missing a period. -On line 322, 'in case of' should be 'in the case of' minor suggestions for appendix: -Section 4 of the experiments refer to 'badges' of experiments, this should probably be 'batches' -There is a space missing in the description of the second set of experiments -Tables 5,6, the meaning of, e.g., 'k in 1000', is not clear. -In Figure 1, for logistic regression, it is very difficult to see the line for theano. Originality ----------- The the originality of the work is to show that a combination of two well-known ideas, Ricci calculus and automatic differentiation, can result in faster, smaller computational graphs that can compute higher order derivatives. Significance ------------ The core idea here has much potential for the machine learning community at large, but the experiments seem more like preliminary results and could be much stronger. Post-rebuttal update =============================== While the rebuttal does clear up certain questions, the paper still needs work on its empirical motivation. Though I believe this work has great potential, my assessment of the paper remains the same for the following primary reason: Any notion of including additional experiments (which is also brought up by another reviewer) to better motivate the work is completely absent in the rebuttal. While the authors argue that their synthetic experiments on three toy problems are enough, they are quite far removed from realistic problems where autodifferentiation with higher order methods would actually be useful. If the authors are going to evaluate on problems that are trivial to solve with typical second order approaches, they need to compare with the typical baseline approach and not just an auto-diff strawman that most people wouldn't use to solve the problem. In further revisions, I would suggest that the authors seriously consider the following: 1) Formally present the cost of creating the computational graph. While it may seem insignificant to you, the reality is that many second order methods are derived and implemented directly, and not with auto-diff, and so this is the overhead that matters to many people. 2) Include the exact experimental setup for generating the benchmarks. Running time *can* in fact depend on whether the data is synethetic or not; oftentimes real data is sparse or structured, which is often exploited when people implement second order methods. The auto-diff runtime may not be so great if it doesn't exploit various cost-saving structural shortcuts that are common to use in practice. 3) Alternatively, instead of comparing to problems which are trivial to solve without auto-diff, evaluate in a setting where people actually use second order auto-differentiation (e.g. double backprop).